# Seroprevalence of anti-SARS-CoV-2 IgG antibodies in the staff of a public school system in the midwestern United States

Lilah Lopez[1], Thao Nguyen[1], Graham Weber[1], Katlyn Kleimola[2], Megan Bereda[3], Yiling Liu[4,5], Emma K. Accorsi[6], Steven J. Skates[4,5], John P. Santa Maria, Jr.[7], Kendal R. Smith[1‡]*, Mark Kalinich[5‡]*

1 Lake Central School Corporation, Saint John, IN, United States of America, 2 City of Evanston, Evanston, IL, United States of America, 3 Independent Researcher, Cleveland, OH, United States of America, 4 Biostatistics Center, Massachusetts General Hospital, Boston, MA, United States of America, 5 Harvard Medical School, Boston, MA, United States of America, 6 Department of Epidemiology, Harvard T.H. Chan School of Public Health, Boston, MA, United States of America, 7 Novartis Institutes for Biomedical Research, Cambridge, MA, United States of America

☉ These authors contributed equally to this work.
‡ These authors also contributed equally to this work.
* mark_kalinich@hms.harvard.edu (MK); ksmith@lcscmail.com (KRS)

**Data Availability Statement:** All relevant data are within the manuscript and its Supporting Information files.

## Abstract

Since March 2020, the United States has lost over 580,000 lives to severe acute respiratory syndrome coronavirus 2 (SARS-CoV-2), which causes COVID-19. A growing body of literature describes population-level SARS-CoV-2 exposure, but studies of antibody seroprevalence within school systems are critically lacking, hampering evidence-based discussions on school reopenings. The Lake Central School Corporation (LCSC), a public school system in suburban Indiana, USA, assessed SARS-CoV-2 seroprevalence in its staff and identified correlations between seropositivity and subjective histories and demographics. This study is a cross-sectional, population-based analysis of the seroprevalence of SARS-CoV-2 in LCSC staff measured in July 2020. We tested for seroprevalence with the Abbott Alinity™ SARS-CoV-2 IgG antibody test. The primary outcome was the total seroprevalence of SARS-CoV-2, and secondary outcomes included trends of antibody presence in relation to baseline attributes. 753 participants representative of the staff at large were enrolled. 22 participants (2.9%, 95% CI: 1.8% - 4.4%) tested positive for SARS-CoV-2 antibodies. Correcting for test performance parameters, the seroprevalence is estimated at 1.7% (90% Credible Interval: 0.27% - 3.3%). Multivariable logistic regression including mask wearing, travel history, symptom history, and contact history revealed a 48-fold increase in the odds of seropositivity if an individual previously tested positive for COVID-19 (OR: 48, 95% CI: 4–600). Amongst individuals with no previous positive test, exposure to a person diagnosed with COVID-19 increased the odds of seropositivity by 7-fold (OR: 7.2, 95% CI: 2.6–19). Assuming the presence of antibodies is associated with immunity against SARS-CoV-2 infection, these results demonstrate a broad lack of herd immunity amongst the school corporation's staff irrespective of employment role or location. Protective measures like contact tracing, face coverings, and social distancing are therefore vital to maintaining the safety of both students and staff as the school year progresses.

**Funding:** K.K. was supported by the City of Evanston. Y.L. and S.S. were supported by the MGH Biostatistics Center and Harvard Medical School. E.K.A. was supported by the Harvard T.H. Chan School of Public Health. J.S.M was supported by the Novartis Institute for Biomedical Research. K.R.S. was supported by the Lake Central School Corporation. M.K. was supported by Harvard Medical School and NIH grants T32GM007753 and F30 CA224588. The funders had no role in study design, data collection and analysis, decision to publish, or preparation of the manuscript.

**Competing interests:** One author was a full-time employee of Novartis Institutes for Biomedical Research (NIBR) during the preparation of this work (John P. Santa Maria Jr.). NIBR had no role in study design, data collection and analysis, decision to publish, or preparation of the manuscript. This does not alter our adherence to PLOS ONE policies on sharing data and materials.

## Introduction

Over 3 million people around the world have lost their lives from severe acute respiratory syndrome coronavirus 2 (SARS-CoV-2, or COVID-19) as of May 2021. The United States alone has reported over 580,000 deaths [1]. A critical bottleneck in containing the virus is understanding its transmission dynamics. Although much effort has been expended on population-level seroprevalence surveys, scant data exist to understand COVID-19's potential effect on the US public school system's students and staff. Existing data are sobering: ten days into its reopening, an Israeli high school experienced a major outbreak [2]. Work on the pediatric transmission of COVID-19 are conflicting; in multiple studies conducted in Southwest Germany, Ireland, and Northern France, children under 10 were found to have had little effect on the spread of the virus, while a study in Chile claimed that elementary students were more likely to contract the virus relative to secondary students [3–6]. Here, we determine the seroprevalence of COVID-19 in the Lake Central School Corporation (LCSC), a public school system located in suburban Indiana, US. The LCSC consists of 1 high school (grades 9–12; 3,261 students), 3 middle schools (grades 5–8; 2,859 students), and 6 elementary schools (grades kindergarten-4, 3,467 students), and transitioned to fully virtual instruction on March 16[th] 2020 [7]. Although there is enormous heterogeneity within the US public school system, LCSC staff's demographics are broadly representative of national statistics. The LCSC has 16.6 students per teacher, similar to the United States average of 16 students per teacher [8, 9]. The median age of public school employees in the United States is 41, while the median age of the Lake Central employee population lies at 48 [10, 11]. Notably, previous reports have established that advanced age is highly associated with COVID-19 hospitalization, further underscoring the threat of morbidity and mortality within this community relative to other school districts [12]. Given the lack of specificity of COVID-19 symptoms and that mild and asymptomatic cases of COVID-19 may go undocumented, antibody-based seroprevalance studies are required to estimate population-level exposure to SARS-CoV-2, although it should be noted that immunity to SARS-CoV-2 via such antibodies has yet to be firmly established [13].

## Materials and methods

### Study design and participants

This cross-sectional, population-based analysis of the seroprevalence of anti-SARS-CoV-2 IgG enrolled participants over a five day period in July of 2020 as part of the LCSC staff's annual wellness check. Individuals were eligible to participate if they were 18 years of age or older and were employees of the Lake Central school corporation during the 2018–2019 or 2019–2020 school years.

After approval from the Community Healthcare System Central Institutional Review Board (CHS CIRB #07–02), participants were contacted through their respective LCSC email account as well as by voice message from the LCSC superintendent informing them of the opportunity to participate. These communications provided information regarding the study and a link in the email allowed participants to schedule their testing date. A second email containing a video promoting the study was sent to staff and shared on LCSC social media accounts. There was no cost to participate in testing. Once registered, each participant completed a data questionnaire containing questions about sociodemographic characteristics including self-identified gender, employment factors, and activities that can increase the risk of having COVID-19, as well as informed written consent (S1 Text).

During the five days of data collection, the study team was present at the testing site to assist with distributing additional consent forms and data questionnaires. Participants were required

to wear a mask upon entering the building and hand sanitizer was available at each station. Stations were six feet apart, and a member of the LCSC staff used cleaning wipes to disinfect each station between participants. An independent third party vendor, Franciscan WorkingWell, performed venipuncture using standard procedure for the antibody tests, and obtained a 5 mL blood sample for the antibody test and a 6 mL blood sample for blood typing.

The primary outcome was the total seroprevalence of anti-SARS-CoV-2 IgG antibodies within staff. The secondary outcomes were the changes in odds of seropositivity associated with baseline demographics and COVID-19 related factors including mask use, a self-reported history of contact with a known COVID-19 positive person or persons, and a previous positive COVID-19 test (PCR or antibody-based).

## Laboratory analysis

Seroprevalence was determined utilizing the commercially available Abbott Alinity™ SARS-CoV-2 IgG antibody test, with a reported 100% sensitivity (34/34, 95% CI: 89.7%-100%) and 99% specificity (99/100, 95% CI: 94.5%-100%) in detecting anti-SARS-CoV-2 IgG antibodies [14]. This test provides a binary, present/not present result.

## Statistical analysis

Seroprevalence among LCSC staff was calculated as the proportion of staff members who received a positive antibody test out of the total staff tested. Confidence intervals were first estimated by generating a binomial confidence interval with the statsmodel package in Python v3.7. The seroprevalence was then corrected for uncertainties in the test sensitivity and specificity using previously established statistical approaches [15]. Relative risks (risk ratios) were calculated to identify associations between seropositivity and gender, BMI, blood type, contact history, symptom history, travel history, role in school corporation, school where employed, extracurricular role (coach, club sponsor), mask history, previous positive COVID-19 test, and age on 7/20/20, the last day of sample collection. Employment role was separated into two groups: those departments that worked in the school during the Indiana stay-at-home order (maintenance, technology, and administration) and those that did not (all others). Continuous variables (BMI, age) were binarized by segmenting the variable by its median value. Bonferroni correction was used to correct for multiple testing on the univariate analyses. To determine if having a history of mask-wearing reduced seropositivity, we constructed a causal diagram based on existing literature and first-hand knowledge of the Lake Central school system and a priori identified gender, age, travel history (as a proxy for risk taking behavior), school type, and role at LCSC as potential confounders to control for in the multivariable logistic regression (S1 Fig) [6, 16–19]. Finally, backward-stepwise feature elimination was used to identify the features most predictive of seropositivity in a multivariable logistic regression starting with the thirteen categories queried in the survey (threshold p-value of 0.05). After the model was identified for which all factors were statistically significant, three factors (mask history, travel history, and symptom history) which are understood based on external studies to have an impact on infectivity were reintroduced to this reduced model [14–16].

One factor, having a previous positive COVID-19 test in four individuals, was overwhelmingly predictive for seropositivity. The regression was then re-fitted on the vast majority of individuals without a previous positive test to estimate associations of features with seropositivity within this group. For continuous variables such as age and BMI, missing data were either removed (univariate analyses) or replaced with the median of existing data for that variable (multivariable analysis). Missing data for categorical variables were either removed (univariate analyses) or replaced with the mode of that variable (multivariable analysis). A

sensitivity analysis which removed incomplete observations from the multivariable analyses (rather than replacing the missing values) was also performed. Logistic regressions were calculated using the scikit-learn Python package.

## Results and discussion

Of the eligible 1261 staff members, 753 (60%) participated in the study. The LCSC staff comprises 1060 (84%) women; the participation group had 635 (85%) women; age demographics for the study population compared to that of the total staff were similarly representative (Table 1).

22 individuals tested positive for anti-SARS-CoV-2 IgG antibodies, representing 2.9% seroprevalence (95% CI: 1.8% - 4.4%). After correcting for the reported sensitivity and specificity of the IgG antibody test, the seroprevalence is estimated as 1.7% (90% Credible Interval: 0.27% - 3.3%) [20].

In the univariate analyses, having a previous positive COVID-19 test (RR 29.5, 95% CI: 14.3–60.4, p<0.0001) or contact with a COVID-19 case (RR 6.86, 95% CI: 3.04–15.5), p<0.0001) were found to be statistically significantly associated with seropositivity, after Bonferroni correction (Table 2).

No causal relationship between self-reported mask-wearing history and seropositivity could be identified, after controlling for six confounders available in our data set, identified using a causal diagram (S1 Table) (RR 0.63, 95% CI 0.16–4.3). This result was maintained when missing data were replaced, rather than excluded (S2 Table). In the multivariable model employing backward-stepwise feature elimination, a previous positive COVID-19 test (OR 48 95% CI 3.9–600, p = 0.003) or contact with a COVID-19 case (OR 5.6, 95% CI 2.1–15.1, p = 0.001) were found to be the most predictive factors for seropositivity (S3 Table); similar results were observed when missing data were replaced, rather than excluded (S4 Table). To interrogate the specific interaction between contact history and seropositivity, the dataset was subsetted to the vast majority of individuals without a previous positive COVID-19 test and complete data (n = 727). In this population, a logistic regression analysis including the predictors mask wearing, travel history, and symptom history, a positive contact history conferred a 7-fold increase in the odds of seropositivity (p<0.0001) (Table 3). The odds increased from 1:54 without such a contact to 1:7 with contact—or the probability of seropositivity increased from 1.8% to 12%. Similar results were observed when replacing, rather than excluding, missing data (S5 Table).

**Table 1. Characteristics of study population and total staff population.**

| Characteristic | Study Population | | Total Staff Population | |
|---|---|---|---|---|
| | *n* | *%* | *n* | *%* |
| Gender* | | | | |
| Female | 635 | 84.7 | 1060 | 84.1 |
| Age** | | | | |
| 18–35 | 153 | 20.3 | 261 | 20.7 |
| 36–50 | 287 | 38.2 | 483 | 38.3 |
| 51–65 | 282 | 37.5 | 455 | 36.1 |
| 66+ | 30 | 4.0 | 62 | 4.9 |

Note: Three participants did not share their gender and one did not share their age.

*Chi-square value 0.0025, p-value 0.960.

**Chi-square value 1.308, p-value 0.727.

**Table 2. Univariate analysis.**

| Characteristic | Relative Risk | 95% Confidence Interval | P |
|---|---|---|---|
| Covid Status (Self-reported Previous Positive Test) | 29.5 | (14.3, 60.4) | <0.0001 |
| Contact History (Yes) | 6.86 | (3.04, 15.5) | <0.0001 |
| Club Sponsor | 3.10 | (1.22, 7.84) | 0.0258 |
| Symptom History (Yes) | 2.31 | (0.990, 5.42) | 0.0577 |
| BMI (Binary) | 2.17 | (0.884, 5.25) | 0.0867 |
| Was Working at school (in the summer) | 0.196 | (0.027, 1.45) | 0.0979 |
| Sports Coach | 1.96 | (0.591, 6.49) | 0.225 |
| Gender (Female) | 0.616 | (0.232, 1.64) | 0.362 |
| Mask Wearing (Yes) | 0.721 | (0.172, 3.03) | 0.654 |
| Age (Binary) | 1.28 | (0.559, 2.92) | 0.667 |
| Travel History (Yes) | 0.744 | (0.255, 2.17) | 0.798 |
| Elementary School | *ref* | - | |
| Middle School | 2.25 | (0.813, 6.20) | 0.117 |
| High School | 1.15 | (0.357, 3.73) | 1.00 |
| Other School | 1.10 | (0.226, 5.35) | 1.00 |
| Blood Type (O+) | *ref* | - | |
| Blood Type (B-) | 3.28 | (0.475, 22.6) | 0.286 |
| Blood Type (B+) | 0.336 | (0.044, 2.58) | 0.468 |
| Blood Type (A+) | 0.758 | (0.293, 1.96) | 0.628 |
| Blood Type (A-) | 0.444 | (0.058, 3.40) | 0.696 |

AB+, AB-, and O- had no positive cases and therefore have no RR estimate.

This study provides the necessary baseline for future longitudinal monitoring of COVID-19 transmission through a representative US public school system. With a demographically representative 60% staff participation rate, these data offer an unprecedented view into the seroprevalence of the target population of a Midwest US public school system. The estimated 1.7% seroprevalence of COVID-19 antibodies among LCSC staff is far below that reported in metropolitan areas throughout the US at the time of testing, underscoring the large and continued risk of COVID-19 infection within this community upon exposure [20]. As a point of comparison, Lake County, Indiana, the county within which the LCSC resides, had 4,985 reported cases of COVID-19, representing 1.0% of the county's population [21].

Although the effect of mask wearing on seropositivity, utilizing a causal diagram to identify potential confounders, did not achieve statistical significance, this result should not be interpreted as a lack of efficacy in the use of masks given the overwhelming quality and quantity of

**Table 3. Multivariable model.**

| Effect | Odds Ratio | 95% CI | | p |
|---|---|---|---|---|
| | | LL | UL | |
| Intercept | 0.025 | 0.004 | 0.092 | <0.001 |
| Contact History | 7.2 | 2.6 | 19. | <0.001 |
| Symptom History | 2.0 | 0.72 | 5.1 | 0.16 |
| Travel History | 0.47 | 0.11 | 1.5 | 0.25 |
| Mask History | 0.68 | 0.18 | 4.5 | 0.63 |

*Note*: CI = confidence interval; *LL* = CI lower limit; *UL* = CI upper limit.

data supporting their continued use, and is likely due to lack of power [22]. Both univariate and multivariable analyses demonstrate that a previous positive COVID-19 test or a history of contact with a COVID-19 patient were associated with seropositivity. The 7-fold increase in odds of seropositivity for individuals with a positive contact history is especially noteworthy amongst the vast majority (n = 727) of the individuals without a previous positive COVID-19 test and complete data. Although these relationships are associational, the combination of low staff seroprevalence and the strong positive association of seropositivity with contact history highlights a high risk group for which the importance of aggressive contact tracing would be efficient and which would minimize the transmission of COVID-19, as well as continued protective procedures during school hours such as mask and face shield wearing, and social distancing [23].

This study has several limitations. Although data were collected from staff employed at 11 different sites (10 schools and the transportation facility), only the LCSC was involved in the study, limiting the generalizability of this work. Neither ethnicity nor income data were collected, precluding analysis of these variables' previously demonstrated associations with COVID-19 positivity [24]. We rely on study participants to self-report variables, excluding antibody status and blood type, via questionnaire. Some questions may be insufficiently granular, such as the binary mask wearing variable, and participants may also make errors filling out the questionnaire. Insufficient granularity is likely to make groups defined by the variable more similar to each other, while reporting errors are likely to be random and unrelated to the outcome, antibody status. Thereby both will be expected to create bias towards the null, meaning that the results reported in this study are conservative. The low baseline seroprevalence (22 positive tests in the 753-person cohort) prevents the identification of more subtle, but potentially real, associations among the collected variables and seropositivity. Additionally, the lack of county-level seroprevalence data prevents comparison to the broader population outside of the school district. Given that not all staff elected to participate in the study, some level of volunteer bias is possible. The high participation rate (60.0%), highly representative sample population (Table 1) and the wide availability of COVID-19 testing outside of this study, however, all limit the potential for volunteer bias to drive the observed results [25]. Finally, data was not collected on students in the corporation, preventing the investigation of a potential link between staff or student positivity and transmission within or between these two co-exposed groups.

The reopening of US schools poses a potential risk of COVID-19 transmission to both staff and students. The low pre-opening seroprevalence, in combination with the advanced age of a significant fraction of the LCSC staff, may increase the number and severity of cases should an outbreak occur within this school system, or other school systems sharing its demographic characteristics. Teachers should consider deploying novel teaching strategies that limit the amount of non-distanced interactions. Administrators and legislators should allocate the required resources to implement and maintain robust personal protective measures as recommended by the CDC, including contact-tracing, masks, face shields, and social distancing to protect the lives of both the children of the school district and those charged with educating them [26].

## Supporting information

**S1 Dataset. De-identified LCSC COVID dataset.**
(CSV)

**S1 Fig. Construction of a causal diagram to identify confounders of the relationship between mask wearing and seropositivity.**
(DOCX)

**S1 Table. Logistic regression results for the relationship between mask wearing history and seropositivity, adjusting for potential confounders (missing data excluded).** (DOCX)

**S2 Table. Logistic regression results for the relationship between mask wearing history and seropositivity, adjusting for potential confounders (missing data replaced).** (DOCX)

**S3 Table. Stepwise backwards feature elimination regression results (missing data excluded).** (DOCX)

**S4 Table. Stepwise backwards feature elimination regression results (missing data replaced).** (DOCX)

**S5 Table. Stepwise backwards feature elimination regression results, previously COVID + patients excluded (missing data replaced).** (DOCX)

**S1 Text. Data collection for seroprevalence study of COVID 19 in Lake Central Staff.** (DOCX)

## Acknowledgments

We thank all of the LCSC staff members who participated in this study. We also thank Jana L. Lacera, Director, IRB/Bio-Ethics, who graciously volunteered both her and her organization's resources; the nurses from Franciscan WorkingWell for collecting participant samples; Dr. Marc Lipsitch for his invaluable guidance, feedback and encouragement; and the LCSC for enabling this work.

## Author Contributions

**Conceptualization:** Lilah Lopez, Graham Weber, Megan Bereda, Kendal R. Smith, Mark Kalinich.

**Data curation:** Lilah Lopez, Thao Nguyen, Graham Weber, John P. Santa Maria, Jr., Kendal R. Smith, Mark Kalinich.

**Formal analysis:** Lilah Lopez, Thao Nguyen, Graham Weber, Katlyn Kleimola, Yiling Liu, Emma K. Accorsi, Steven J. Skates, John P. Santa Maria, Jr., Kendal R. Smith, Mark Kalinich.

**Funding acquisition:** Kendal R. Smith.

**Investigation:** Lilah Lopez, Thao Nguyen, Graham Weber, John P. Santa Maria, Jr., Kendal R. Smith, Mark Kalinich.

**Methodology:** Thao Nguyen, Graham Weber, Katlyn Kleimola, Yiling Liu, Emma K. Accorsi, Steven J. Skates, John P. Santa Maria, Jr., Mark Kalinich.

**Project administration:** Lilah Lopez, Graham Weber, Megan Bereda, Emma K. Accorsi, Steven J. Skates, John P. Santa Maria, Jr., Kendal R. Smith, Mark Kalinich.

**Resources:** Lilah Lopez, Kendal R. Smith.

**Supervision:** Katlyn Kleimola, Megan Bereda, Emma K. Accorsi, Steven J. Skates, John P. Santa Maria, Jr., Kendal R. Smith, Mark Kalinich.

**Validation:** Yiling Liu, Emma K. Accorsi, Steven J. Skates, John P. Santa Maria, Jr., Mark Kalinich.

**Visualization:** Thao Nguyen.

**Writing – original draft:** Lilah Lopez, Thao Nguyen, Graham Weber, Katlyn Kleimola, Megan Bereda, Emma K. Accorsi, Steven J. Skates, John P. Santa Maria, Jr., Kendal R. Smith, Mark Kalinich.

**Writing – review & editing:** Lilah Lopez, Thao Nguyen, Graham Weber, Katlyn Kleimola, Megan Bereda, Emma K. Accorsi, Steven J. Skates, John P. Santa Maria, Jr., Kendal R. Smith, Mark Kalinich.

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
