## [Decision Letter · Decision Letter 0]

30 Mar 2021

PONE-D-20-38006

Seroprevalence of anti-SARS-CoV-2 IgG Antibodies in the Staff of a Public School System in the Midwestern United States

PLOS ONE

Dear Dr. Kalinich,

Thank you for submitting your manuscript to PLOS ONE. After careful consideration, we feel that it has merit but does not fully meet PLOS ONE’s publication criteria as it currently stands. Therefore, we invite you to submit a revised version of the manuscript that addresses the points raised during the review process.

Your manuscript was reviewed by one expert in the field. Unfortunately, many potential reviewers did not accept the review invitation. Nevertheless, this reviewer identified all major points that require your attention. Please carefully consider the attached comments and provide thorough responses.

We look forward to receiving your revised manuscript.

Kind regards,

Yury E Khudyakov, PhD

Academic Editor

PLOS ONE

Journal Requirements:

'The author(s) received no specific funding for this work. The Lake Central School Corporation funded the seroprevalence testing. '

We note that one or more of the authors are employed by commercial companies: Novartis, Independent Researcher.

a. Please provide an amended Funding Statement declaring these commercial affiliations, as well as a statement regarding the Role of Funders in your study. If the funding organization did not play a role in the study design, data collection and analysis, decision to publish, or preparation of the manuscript and only provided financial support in the form of authors' salaries and/or research materials, please review your statements relating to the author contributions, and ensure you have specifically and accurately indicated the role(s) that these authors had in your study. You can update author roles in the Author Contributions section of the online submission form.

b. Please also provide an updated Competing Interests Statement declaring these commercial affiliations along with any other relevant declarations relating to employment, consultancy, patents, products in development, or marketed products, etc.  

4. Please include your tables as part of your main manuscript and remove the individual files. Please note that supplementary tables should remain as separate "supporting information" files.

5. Please include captions for your Supporting Information files at the end of your manuscript, and update any in-text citations to match accordingly. Please see our Supporting Information guidelines for more information: http://journals.plos.org/plosone/s/supporting-information

Reviewers' comments:

Reviewer's Responses to Questions

**Comments to the Author**

1. Is the manuscript technically sound, and do the data support the conclusions?

Reviewer #1: No

2. Has the statistical analysis been performed appropriately and rigorously? 

Reviewer #1: No

3. Have the authors made all data underlying the findings in their manuscript fully available?

Reviewer #1: No

4. Is the manuscript presented in an intelligible fashion and written in standard English?

Reviewer #1: Yes

5. Review Comments to the Author

Reviewer #1: This article examines the seroprevalence among a staff of a public school in the Midwestern in the USA. The aim is to get an idea of the seroprevalence and the associated risk factors to determine whether it makes sense to reopen the schools.

The question is indeed interesting and interests a much wider audience than the United States alone. However, the paper is sorely lacking in precision and rigor at all levels of its structure:

1. Introduction

We do not know what is the "student" population concerned by this study, is it a primary school, a high school ...? Knowing that there are disparities in the circulation of SARS-CoV-2 according to the age of the children, it is absolutely necessary to mention it, to discuss it at a minimum if it cannot be taken into account.

2. Methods

a. Regarding this question of the school's student population, it would have been interesting to note the seroprevalence among them as well, to determine whether or not it was higher than among the staff. An adapted questionnaire would have made it possible to highlight the relevance of the barrier measures put in place.

b. It is announced that the variable "contact with a positive person" will be analyzed, without specifying whether this contact concerns a colleague or a relative. It would be desirable to specify and take into account these two modes of contact.

c. You announce that you have replaced the missing data with the median or the mode depending on the variable. You thus create a huge bias in your data, especially when you know that they relate, for example, to age and BMI, which have a major influence on the attack by SARS-CoV-2. Missing data should be excluded.

d. Where is the methodological part concerning the DAG? There are a huge number of methods to build a DAG, which one have you used? With which package and which assumptions ?

3. Results

a. You describe a fragmented participation in your study. What about the biases associated with this participation rate?

b. Where is the DAG produced after all? We only see the DAG of the tested hypotheses. Again, what about the package used?

4. Discussion

To rule on the reopening of schools, it would have been necessary to determine whether the school staff was more affected than the general population, taking into account the participation bias. This issue of bias is not addressed.

To judge the relevance of maintaining the barrier measures within the school, it would have been necessary to compare the seroprevalence of the staff both with that of the general population under these conditions of bias but also with the seroprevalence of the student population of the school and assess the adherence of these populations to barrier measures. Without these comparisons no conclusion can be given on these points.

6. PLOS authors have the option to publish the peer review history of their article (what does this mean?). If published, this will include your full peer review and any attached files.

Reviewer #1: No

---

## [Author Response · Author response to Decision Letter 0]

12 May 2021

Journal Requirements:

We have modified our draft to conform to the above guidance. 

We agree that we should share as much of the data as is feasible for the broader scientific community’s research efforts in combating COVID-19. We have attached the IRB determination after sharing the above request, which has permitted us to share the following variables: 

● Age by decile

● Gender

● BMI

● Symptom History

● Travel History

● COVID Status

● Mask History

● School type (elementary, middle, high school) 

● Contact History

● Antibody test result

● Blood type

These variables will enable the broader scientific community to recapitulate our key findings while maintaining the anonymity of our participants, and are included as S1 Dataset: De-Identified LCSC COVID Dataset.

3.Thank you for stating the following in the Financial Disclosure section:

'The author(s) received no specific funding for this work. The Lake Central School Corporation funded the seroprevalence testing. '

We note that one or more of the authors are employed by commercial companies: Novartis, Independent Researcher.

a. Please provide an amended Funding Statement declaring these commercial affiliations, as well as a statement regarding the Role of Funders in your study. If the funding organization did not play a role in the study design, data collection and analysis, decision to publish, or preparation of the manuscript and only provided financial support in the form of authors' salaries and/or research materials, please review your statements relating to the author contributions, and ensure you have specifically and accurately indicated the role(s) that these authors had in your study. You can update author roles in the Author Contributions section of the online submission form.

We thank the editor for suggesting this clarification of the role of these funding sources and have amended the funding statement to the following: 

“K.K. was supported by the City of Evanston. Y.L. and S.S. were supported by the MGH Biostatistics Center and Harvard Medical School. E.K.A. was supported by the Harvard T.H. Chan School of Public Health. J.S.M was supported by the Novartis Institute for Biomedical Research. K.R.S. was supported by the Lake Central School Corporation. M.K. was supported by Harvard Medical School and NIH grants T32GM007753 and F30 CA224588. The funders had no role in study design, data collection and analysis, decision to publish, or preparation of the manuscript.”

At the time of writing, author MB was unemployed; how would you recommend we refer to her affiliation (currently listed as ‘Independent Researcher’? 

b. Please also provide an updated Competing Interests Statement declaring these commercial affiliations along with any other relevant declarations relating to employment, consultancy, patents, products in development, or marketed products, etc. 

Given that the funders had no role in the study design, data collection and analysis, decision to publish, or preparation of the manuscript, nor at any time did anything that interferes with, or could reasonably be perceived as interfering with, the full and objective presentation, peer review, editorial decision-making, or publication of research or non-research articles submitted to one of the journals, we do not believe any author has a competing interest to declare, outside of their existing commercial affiliations. We have modified the competing interest statement to the following: 

“One author was a full-time employee of Novartis Institutes for Biomedical Research (NIBR) during the preparation of this work (John P. Santa Maria Jr.). NIBR had no role in study design, data collection and analysis, decision to publish, or preparation of the manuscript. This does not alter our adherence to PLOS ONE policies on sharing data and materials.”

We confirm that these commercial affiliations do not alter our adherence to PLOS ONE policies. Any restrictions on data sharing stem from the IRB’s determination as above. 

We have included both an updated Funding Statement and Competing Interest Statement in the body of the cover letter. 

4. Please include your tables as part of your main manuscript and remove the individual files. Please note that supplementary tables should remain as separate "supporting information" files.

We have added our tables as part of the main manuscript and removed the individual files. 

5. Please include captions for your Supporting Information files at the end of your manuscript, and update any in-text citations to match accordingly. Please see our Supporting Information guidelines for more information: http://journals.plos.org/plosone/s/supporting-information

We have added captions for the supporting information files at the end of our manuscript file, and have updated the in-text citations to match accordingly. 

Comments to the Author

Reviewer #1: This article examines the seroprevalence among a staff of a public school in the Midwestern in the USA. The aim is to get an idea of the seroprevalence and the associated risk factors to determine whether it makes sense to reopen the schools.

The question is indeed interesting and interests a much wider audience than the United States alone. However, the paper is sorely lacking in precision and rigor at all levels of its structure:

We greatly appreciate the reviewer’s time and input on the study, and their interest in this important subject area. As the reviewer pointed out, our study aims to fill a large gap in the current COVID-19 literature by identifying risk factors for seropositivity within a school system in order to better understand the role of schools in SARS-CoV-2 transmission. 

1. Introduction 

We do not know what is the "student" population concerned by this study, is it a primary school, a high school ...? Knowing that there are disparities in the circulation of SARS-CoV-2 according to the age of the children, it is absolutely necessary to mention it, to discuss it at a minimum if it cannot be taken into account.

We thank the reviewer for pointing out the lack of an LCSC student population profile. We have added the following text to the introduction: 

“The LCSC consists of 1 high school (grades 9-12; 3,261 students), 3 middle schools (grades 5-8; 2,859 students), and 6 elementary schools (grades kindergarten-4, 3,467 students), and transitioned to fully virtual instruction on March 16th 2020.” 

2. Methods 

a.Regarding this question of the school's student population, it would have been interesting to note the seroprevalence among them as well, to determine whether or not it was higher than among the staff. An adapted questionnaire would have made it possible to highlight the relevance of the barrier measures put in place.

We thank the reviewer for this thoughtful comment, and agree that such data could be of great scientific value. With respect to barrier measures: starting on March 16th, 2020, the LCSC transitioned entirely to virtual teaching. Barrier measures were therefore not implemented at LCSC between the start of the pandemic and the time of testing. We have added the following text to the manuscript to clarify this point (new text bolded): 

“The LCSC consists of 1 high school (grades 9-12; 3,261 students), 3 middle schools (grades 5-8; 2,859 students), and 6 elementary schools (grades kindergarten-4, 3,467 students), and transitioned to fully virtual instruction on March 16th 2020.” 

With respect to the seroprevalence of the LCSC student population: unfortunately, we were unable to collect these data for multiple reasons. Funding was unavailable for surveilling the student population from the LCSC, as the staff’s insurer paid for the cost of the test. Additionally, our IRB had ethical concerns about LCSC students analyzing fellow students’ data (a majority pediatric population) at the level of granularity that was to be performed even if we were to successfully raise funding. We have modified the text to reflect that this analysis was a component of the LCSC’s annual staff annual wellness check, as below (new text bolded): 

“This cross-sectional, population-based analysis of the seroprevalence of anti-SARS-CoV-2 IgG enrolled participants over a five day period in July of 2020 as part of the LCSC staff’s annual wellness check.” 

b.It is announced that the variable "contact with a positive person" will be analyzed, without specifying whether this contact concerns a colleague or a relative. It would be desirable to specify and take into account these two modes of contact.

We agree with the reviewer that if school had been in session physically, this information would be crucial to assisting in deconvolving home versus school exposure, and have incorporated this feedback into a subsequent survey sent to the staff for a separate project. Given that the LCSC transitioned to fully-virtual instruction on 3/16/2020, the staff (with the exception of the administrative and janitorial staff), exposure from colleagues was negligible. Additionally, the LCSC had no method or policy of informing staff of exposure to a positive staff member until after the time of the LCSC annual wellness check. We have updated the manuscript Methods section (pertinent section bolded) to provide clarification:

“The primary outcome was the total seroprevalence of anti-SARS-CoV-2 IgG antibodies within staff. The secondary outcomes were the changes in odds of seropositivity associated with baseline demographics and COVID-19 related factors including mask use, a self-reported history of contact with a known COVID-19 positive person or persons, and a previous positive COVID-19 test (PCR or antibody-based).”

c. You announce that you have replaced the missing data with the median or the mode depending on the variable. You thus create a huge bias in your data, especially when you know that they relate, for example, to age and BMI, which have a major influence on the attack by SARS-CoV-2. Missing data should be excluded.

We thank the reviewer for this excellent feedback, and agree that a sensitivity analysis that excludes, rather than replaces, missing data for the multivariable analyses is appropriate (the univariable analyses used excluded data in the initial submission). We now report the results for the excluded data throughout the main manuscript, and provide the results from replaced data as supplementary tables.

Reassuringly, both of the resulting multivariable analyses where missing data were excluded maintained qualitatively identical results to our initial results. The causal diagram with excluded data found no statistically significant link between mask-wearing and COVID seropositivity (RR 0.63, 95% CI 0.16-4.3) relative to the replaced data (RR 0.83, 95% CI 0.18 - 3.8). We have modified the text to read: 

No causal relationship between self-reported mask-wearing history and seropositivity could be identified, after controlling for six confounders available in our data set, identified using a causal diagram (S1 Table) (RR 0.63, 95% CI 0.16-4.3). This result was maintained when missing data were replaced, rather than excluded (S2 Table).

Similarly close results were found between the multivariable models employing backward-stepwise feature elimination using replaced or excluded data. We have modified the text to read: 

In the multivariable model employing backward-stepwise feature elimination, a previous positive COVID-19 test (OR 48 95% CI 3.9 - 600, p=0.003) or contact with a COVID-19 case (OR 5.6, 95% CI 2.1 - 15.1, p=0.001) were found to be the most predictive factors for seropositivity (S3 Table); similar results were observed when missing data were replaced, rather than excluded (S4 Table).

While performing the sensitivity analysis for the backward-stepwise feature elimination logistic regression, we realized that we had inadvertently included the results originating from excluded data, rather than replaced data, in our initial submission (which we had performed, but not included, in the initial manuscript). S3 Table and S4 Table have been appropriately updated. 

After subsetting the data to only individuals without a previous positive COVID-19 test, models using either excluded or replaced data yielded similar results. Table 3 has been replaced with the results generated from the excluded data; the results from the replaced data are now in S5 Table. We have updated the text to read: 

In this population, a logistic regression analysis including the predictors mask wearing, travel history, and symptom history, a positive contact history conferred a 7-fold increase in the odds of seropositivity (p<0.0001) (Table 3). The odds increased from 1:54 without such a contact to 1:7 with contact - or the probability of seropositivity increased from 1.8% to 12%. Similar results were observed when replacing, rather than excluding, missing data (S5 Table).

d. Where is the methodological part concerning the DAG? There are a huge number of methods to build a DAG, which one have you used? With which package and which assumptions?

We thank the reviewer for their feedback and for pointing out this opportunity to improve the explanation of our methods. In this paper, we took a causal inference approach to DAGs; we built the DAG using subject-matter knowledge to identify important variables that should be adjusted for in the analysis, which is different from other common statistical approaches to variable selection (Hernán et al., 2002). Therefore, we did not use any specific methods or packages to build the DAG. Based on subject-matter knowledge - including the existing literature and first-hand knowledge of the Lake Central school system - we identified gender, age, travel history (as a proxy for risk taking behavior), school type, and role in the school as variables that could likely affect both mask-wearing behavior and seropositivity, making them potential confounders that must be adjusted for in order to interpret our estimate for the effect of mask-wearing on seropositivity causally.

To make it more clear that we are referring to a causal inference perspective, we have replaced the word directed acyclic graph/DAG with “causal diagram” throughout the manuscript. 

We have updated the main manuscript text to include relevant citations for the variables included on the DAG and to be more explicit that the DAG was generated a priori from subject-matter knowledge:

“To determine if having a history of mask-wearing reduced seropositivity, we constructed a causal diagram based on existing literature and first-hand knowledge of the Lake Central school system and a priori identified gender, age, travel history (as a proxy for risk taking behavior), school type, and role at LCSC as potential confounders to control for in the multivariable logistic regression (S1 Figure).”

We have also updated the S1 Figure caption and text to provide more details on our assumptions:

“S1 Figure: Construction of a Causal Diagram to Identify Confounders of the Relationship Between Mask Wearing and Seropositivity 

This causal diagram is drawn under the null hypothesis of no effect of mask-wearing on seropositivity and shows the relationships between important variables hypothesized to affect both mask-wearing and seropositivity. We adjusted for these potential confounders to identify the effect of mask-wearing on seropositivity. To interpret this estimate causally, we assume that there are no unmeasured confounding variables and that our logistic model is correctly specified; however, it is possible that risk-taking personality may affect seroprevalence in other ways such as not social distancing, but those data weren’t collected and could not be adjusted for in this round of testing.”

3. Results

A. You describe a fragmented participation in your study. What about the biases associated with this participation rate?

We very much agree with the reviewer that biases related to participation could occur and should be discussed. In particular, we recently described potential recruitment-based biases present in serosurveys for SARS-CoV-2 (Accorsi et al., 2021). Firstly, ascertainment bias will occur if the people present for sampling are at lower or higher risk of COVID-19 than average due to the sampling location and time. Since all LCSC staff were approached to participate in this study, ascertainment bias is not an issue. However, not all staff elected to participate in the study and volunteer bias will occur if individuals are more (or less) likely to accept testing because they believe they’ve previously had COVID-19, resulting in estimates of seroprevalence that are too high (or low). We do not believe volunteer bias is driving the results found in this study because (1) we had an overall high participation rate for a SARS-CoV-2 serosurvey (60.0%), (2) the demographics of the study sample are highly representative of the target population (Table 1), (3) the study was performed during the summer of 2020 when COVID-19 test availability in general was much higher, reducing the need to seek testing through a study. 

We have updated discussion of study limitations to include more information on biases arising from imperfect participation:

Given that not all staff elected to participate in the study, some level of volunteer bias is possible. The high participation rate (60.0%), highly representative sample population (Table 1) and the wide availability of COVID-19 testing outside of this study, however, all limit the potential for volunteer bias to drive the observed results.

B. Where is the DAG produced after all? We only see the DAG of the tested hypotheses. Again, what about the package used?

We thank the reviewer for their thorough review of the paper and for bringing up this lack of clarity related to the use of DAGs. As described above, the DAG was created a priori based on subject-matter knowledge, therefore we did not use any software packages to generate it and we only created one DAG (shown in S1 Figure) for the specific hypothesis we wanted to evaluate (i.e., the causal effect of mask-wearing on seropositivity in LCSC teachers). We used the causal diagram to identify possible sources of structural bias, such as confounding, that would need to be addressed in order to interpret the coefficient for mask wearing causally. As above, we have updated the text to improve clarity by instead using the term “causal diagram”, being more explicit that the DAG was generated a priori using subject-matter knowledge, and stating the assumptions required to interpret model estimates causally.

4. Discussion

To rule on the reopening of schools, it would have been necessary to determine whether the school staff was more affected than the general population, taking into account the participation bias. This issue of bias is not addressed.

We agree with the reviewer that determining the seropositivity of the general population would have been crucial for determining the effect of school re-opening on COVID transmission. The main purpose of our work was to understand the baseline seroprevalence of the LCSC staff, and therefore the fraction of the LCSC staff at risk for the sequelae of COVID-19 infection. We have changed the manuscript discussion section to clarify and support this purpose: 

“This study provides the necessary baseline for future longitudinal monitoring of COVID-19 transmission through a representative US public school system.”

“The estimated 1.7% seroprevalence of COVID-19 antibodies among LCSC staff is far below that reported in metropolitan areas throughout the US at the time of testing, underscoring the large and continued risk of COVID-19 infection within this community upon exposure.16”

We thank the reviewer for correctly identifying the lack of discussion about participation bias. We have modified the text to address bias as described above. 

To judge the relevance of maintaining the barrier measures within the school, it would have been necessary to compare the seroprevalence of the staff both with that of the general population under these conditions of bias but also with the seroprevalence of the student population of the school and assess the adherence of these populations to barrier measures. Without these comparisons no conclusion can be given on these points.

The LCSC was fully virtual as of 3/16/2020, and barrier protections were therefore not deployed. We have clarified in the discussion section that such recommendations are a reinforcement on CDC guidance concerning school re-openings: 

“Administrators and legislators should allocate the required resources to implement and maintain robust personal protective measures as recommended by the CDC, including contact-tracing, masks, face shields, and social distancing to protect the lives of both the children of the school district and those charged with educating them.”

References

Accorsi, E. K., Qiu, X., Rumpler, E., Kennedy-Shaffer, L., Kahn, R., Joshi, K., Goldstein, E., Stensrud, M. J., Niehus, R., Cevik, M., & Lipsitch, M. (2021). How to detect and reduce potential sources of biases in studies of SARS-CoV-2 and COVID-19. European Journal of Epidemiology. https://doi.org/10.1007/s10654-021-00727-7

Hernán, M. A., Hernández-Díaz, S., Werler, M. M., & Mitchell, A. A. (2002). Causal knowledge as a prerequisite for confounding evaluation: an application to birth defects epidemiology. American Journal of Epidemiology, 155(2), 176–184.

---

## [Editor Report · Decision Letter 1]

17 May 2021

Seroprevalence of anti-SARS-CoV-2 IgG Antibodies in the Staff of a Public School System in the Midwestern United States

PONE-D-20-38006R1

Dear Dr. Kalinich,

We’re pleased to inform you that your manuscript has been judged scientifically suitable for publication and will be formally accepted for publication once it meets all outstanding technical requirements.

Kind regards,

Yury E Khudyakov, PhD

Academic Editor

PLOS ONE
---

## [Editor Report · Acceptance letter]

31 May 2021

PONE-D-20-38006R1 

Seroprevalence of anti-SARS-CoV-2 IgG antibodies in the staff of a public school system in the midwestern United States 

Dear Dr. Kalinich:

I'm pleased to inform you that your manuscript has been deemed suitable for publication in PLOS ONE. Congratulations! Your manuscript is now with our production department. 

Kind regards, 

on behalf of

Dr. Yury E Khudyakov 

Academic Editor

PLOS ONE